# Hydrogenation of benzoic acid derivatives over Pt/TiO$_2$ under mild conditions

Miao Guo[1], Xiangtao Kong[2], Chunzhi Li[1,3] & Qihua Yang [1✉]

Hydrogenation of benzoic acid (BA) to cyclohexanecarboxylic acid (CCA) has important industrial and academic significance, however, the electron deficient aromatic ring and catalyst poisoning by carboxyl groups make BA hydrogenation a challenging transformation. Herein, we report that Pt/TiO$_2$ is very effective for BA hydrogenation with, to our knowledge, a record TOF of 4490 h$^{-1}$ at 80 °C and 50 bar H$_2$, one order higher than previously reported results. Pt/TiO$_2$ catalysts with electron-deficient and electron-enriched Pt sites are obtained by modifying the electron transfer direction between Pt and TiO$_2$. Electron-deficient Pt sites interact with BA more strongly than electron-rich Pt sites, helping the dissociated H of the carboxyl group to participate in BA hydrogenation, thus enhancing its activity. The wide substrate scope, including bi- and tri-benzoic acids, further demonstrates the high efficiency of Pt/TiO$_2$ for hydrogenation of BA derivatives.

[1] State Key Laboratory of Catalysis, Dalian Institute of Chemical Physics, Chinese Academy of Sciences, Dalian, China. [2] College of Chemistry and Chemical Engineering, Anyang Normal University, Anyang, China. [3] University of Chinese Academy of Sciences, Beijing, China. ✉email: yangqh@dicp.ac.cn

The selective hydrogenation of benzoic acid (BA) or its derivatives has been widely used for the production of fine chemicals, intermediates, and industrial raw materials[1–3]. For example, BA hydrogenation to cyclohexanecarboxylic acid (CCA) is an important step in the production of nylon-6 in industry[4,5]. However, the need to overcome the high resonance energy of the electron-deficient aromatic ring[6] and the catalyst "poisoning" by the carboxyl group[7,8] make BA hydrogenation one of the most challenging transformations. Harsh conditions (100–250 °C, 50–150 bar $H_2$) are typically required in order to obtain high BA conversion, which inevitably causes a decrease in selectivity due to the side reactions of decarboxylation and over-hydrogenation[9,10]. Up to now, various supported metal catalysts (e.g., Pd, Ru, Rh, and Ni) have been used for BA hydrogenation under relatively mild conditions, but the activity is still relatively low[11,12].

Previous results demonstrated that most of the supported metal catalysts are active only with water as solvent under mild conditions and show low or no activity in organic solvents for BA hydrogenation[13]. The typical solvation effect is possibly related with preferential adsorption of aromatic ring on metal surface induced by the interaction of carboxyl groups with $H_2O$ molecules[13,14] and the participation of H* from the dissociated $H_2O$ molecules in the reaction[2,15]. Though water could modify the adsorption mode of substrates, it may also block the metal surface-active sites[16,17]. Taken together, the low $H_2$ solubility in water (e.g., 0.792 mmol $L^{-1}$, 298.15 K, 1 atm $H_2$)[18], water is not a good choice for efficient BA hydrogenation under mild conditions.

Carboxylates tend to strongly adsorb on metal surface, which significantly deteriorates the catalytic activity through so-called "poisoning effects"[19]. In contrast, the adsorption of aromatic ring of BA on metal surface is weak considering that the electron-deficient phenyl ring does not easily bind to the surface unoccupied d-metal orbitals via π-bonds[20,21]. This may be the reason that most metal nanoparticles (NPs) show relatively low activity in BA hydrogenation. Recently, our group reported that the activity of Ru NPs is greatly enhanced in BA hydrogenation by tuning the adsorption mode of BA on Ru surface with phosphine ligands[2]. Therefore, to realize the efficient BA hydrogenation, the supported metal NPs with appropriate adsorption strength toward carboxyl groups and aromatic rings may be a good choice.

Herein, we report that $Pt/TiO_2$ is a highly active and selective catalyst for BA hydrogenation under mild conditions in either organic solvents or water by screening a series of supported metal NPs. To our knowledge, $Pt/TiO_2$ gives a record activity with an apparent TOF up to 4490 $h^{-1}$ at 80 °C and 50 bar $H_2$ in hexane. It was found that electron-deficient Pt site is more active than electron-rich Pt site, which is contributed to the participation of dissociated H from carboxyl groups in BA hydrogenation.

## Results and discussion

**Catalyst screening**. First, commercially available carbon-supported metal NPs were tested in BA hydrogenation (Fig. 1a). Pd/C (5 wt%) and Ru/C (5 wt%) are almost inactive in hexane at 40 °C and 10 bar $H_2$, similar with previous reports[13,22]. To our delight, Pt/C (5 wt%) affords 51% conversion with >99% selectivity to CCA under identical conditions. Inspired by this result, different types of supported Pt catalysts with 2 wt% Pt loading were screened (Fig. 1a) considering that the support with different acid/base or redox properties may influence the catalytic performance of supported metal NPs[23,24]. Pt/MgO, $Pt/CeO_2$, Pt/CN, and $Pt/\gamma-Al_2O_3$ afford <20% BA conversion. $Pt/SiO_2$ and $Pt/ZrO_2$ give moderate BA conversion, 65% and 35%, respectively. $Pt/TiO_2$ affords high BA conversion, 96%, with >99% selectivity

to CCA. The catalyst screen results suggest that the base and acid support, respectively, deteriorate[25] and promote[26] the activity of Pt in BA hydrogenation. $Pd/TiO_2$ and $Ru/TiO_2$ were prepared using the same method with $Pt/TiO_2$. TEM images show that the particle size of Pd and Ru NPs is ca. 4 and 2 nm, respectively (Supplementary Fig. 1). However, $Pd/TiO_2$ and $Ru/TiO_2$ are inactive using hexane as the solvent (Supplementary Table 1), further confirming the advantage of Pt NPs in BA hydrogenation. Even at 25 °C and 1 bar $H_2$, $Pt/TiO_2$ could still afford >99% BA conversion and >99% selectivity to CCA (Table 1). The apparent TOF of $Pt/TiO_2$ was calculated to be 115 $h^{-1}$ at 25 °C, 1 bar $H_2$, and 638 $h^{-1}$ at 40 °C, 10 bar $H_2$. To increase S/C ratio is very important for practical applications. Thus, the BA hydrogenation was performed at S/C as high as 1200 over $Pt/TiO_2$ in the presence of acetic acid to facilitate the dissolution of BA in hexane. Under such harsh conditions, $Pt/TiO_2$ could still afford 90% conversion with an apparent TOF of 4490 $h^{-1}$ at 80 °C and 50 bar $H_2$, an order of magnitude higher than the supported metal NPs ever reported (Supplementary Table 2).

The catalyst screening results suggest that Pt NPs are active for BA hydrogenation in hexane irrespective of the supports, different from Pd and Ru NPs. Density functional theory (DFT) calculation shows that the adsorption energies of BA on Pt (111), Pd (111), and Ru (1000) are, respectively, −1.53, −2.87, and −2.95 eV, showing the stronger adsorption of BA on Ru and Pd than on Pt. The adsorption energies of acetic acid follow the order of Pt (111) < Pd (111) < Ru (1000) in a similar tendency to BA (Fig. 1b). This suggests that the relatively weak adsorption strength of BA on Pt may contribute to the high activity of Pt NPs.

$Pt/TiO_2$ is active in hexane, $H_2O$, cyclohexane, isopropyl alcohol, and EtOH (Table 1 and Supplementary Table 3), showing its wide solvent tolerance. Even using acetic acid as solvent, 68% conversion could still be obtained, showing the high anti-carboxyl poisoning ability of Pt NPs. The product selectivity to CCA is >93% in all the solvents investigated with cyclohexenecarboxylic acid as the only side-product. BA conversion in aprotic and oxygenate-free solvents (e.g., n-hexane) is much higher than that in protic and oxygenate solvents. The decreased hydrogenation rate may be related to the hydrogen bonding of protic solvent with BA, which may hinder the BA adsorption on Pt surface[27]. It is noteworthy to mention that the activity of $Pt/TiO_2$ is much lower in water than in hexane, which is very different from previous results. This can be explained by the two reasons. First, the water may block the active sites of metal NPs due to strong coordination of $H_2O$ (or dissociated OH species) on metal surface[16]. Second, the low solubility of $H_2$ in water may deteriorate the hydrogenation activity[18,28]. In the case of metal NPs (e.g., Pd, Ru) which have strong adsorption strength for BA, the presence of water could weaken the carboxyl adsorption on metal surface by forming H-bonds with BA, which may induce the preferential adsorption of aromatic ring of BA on metal surface to accelerate the activity[13,14]. On the basis of above results, the water could promote the activity of metal NPs with strong adsorption strength for BA but deteriorate the activity of metal NPs with weak adsorption strength for BA as in the case of $Pt/TiO_2$.

**Electronic and geometric structures**. $Pt/TiO_2$-200 and $Pt/TiO_2$-450 were prepared by treatment of $Pt/TiO_2$ under $H_2$ atmosphere at 200 and 450 °C, respectively. The TEM, HRSEM, and HRTEM images of $Pt/TiO_2$, $Pt/TiO_2$-200, and $Pt/TiO_2$-450 showed the uniform distribution of Pt with particle size of 2.9 nm, showing no aggregation of Pt NPs during $H_2$-treatment process (Fig. 2a, b, Supplementary Figs. 2–4). The plane spacing of ca 0.23 nm could

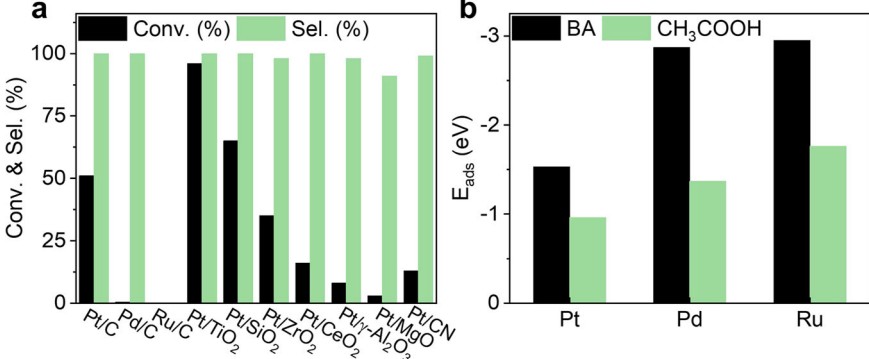

**Fig. 1 Screening the supported metal NPs for BA hydrogenation. a** The catalytic results of supported metal NPs for BA hydrogenation in hexane (40 °C, 10 bar $H_2$, S/C of 700, 2 h). **b** Adsorption energies of BA and acetic acid on Pt (111), Pd (111), and Ru (0001) obtained by DFT calculations. BA benzoic acid.

**Table 1 The catalytic performance of Pt/TiO₂ catalysts in BA hydrogenation[a].**

| Cat. | Solvent | S/C | Conv. (%) | Sel. (%)[b] | TOF (h⁻¹)[c] |
|---|---|---|---|---|---|
| Pt/TiO₂ | Hexane | 250 | >99 | >99 | 638 (2200) |
| | Water | 250 | 84 | >99 | 266 (917) |
| | Acetic acid | 250 | 68 | 96 | 177 (610) |
| Pt/TiO₂[d] | Hexane | 100 | >99 | >99 | 115 (397) |
| Pt/TiO₂[e] | Hexane/acetic acid | 1200 | 90 | 99 | 4490 (15,480) |
| Pt/TiO₂-200 | Hexane | 250 | 59 | >99 | 171 (757) |
| Pt/TiO₂-450 | Hexane | 250 | 10 | 98 | 25 (103) |

[a]Reaction conditions: 40 °C, 10 bar $H_2$, 1 h.
[b]Selectivity to CCA. Only <5% cyclohexenecarboxylic acid was detected as the intermediate during the reaction process.
[c]Apparent TOF is calculated as moles of converted BA per mole of Pt per hour with the conversion <30%. The values in parentheses were the TOF calculated based on Pt dispersion.
[d]25 °C, 1 bar $H_2$, 3 h.
[e]80 °C, 50 bar $H_2$, 1.5 h.

be clearly observed in the HRTEM image of Pt/TiO₂, which is in agreement with the (111) crystal plane of Pt NPs. The XRD patterns Pt/TiO₂ and Pt/TiO₂-450 clearly show the diffraction peaks assigned to rutile TiO₂. No diffraction peaks from Pt NPs could be observed due to the small particle size of Pt (Supplementary Fig. 5). The CO chemisorption results show that Pt dispersion for Pt/TiO₂, Pt/TiO₂-200, and Pt/TiO₂-450 is, respectively, 29.0%, 22.6%, and 24.2% (Table 2). The slight decrease in Pt dispersion of Pt/TiO₂-200 and Pt/TiO₂-450 may be caused by the weak adsorption of CO at the interface of Pt and TiO₂ after $H_2$ treatment[29]. The Pt dispersion was also measured by a HOT method ($H_2$-$O_2$ titration), which affords the similar Pt dispersion tendency to that obtained by CO chemisorption method (Table 2). The higher Pt dispersion obtained from HOT method than from CO chemisorption is possibly due to the $H_2$ spillover effect of Pt/TiO₂[30]. It should be noted that the Pt dispersion of Pt/TiO₂ before and after $H_2$ treatment is comparable, showing that the $H_2$ treatment of Pt/TiO₂ did not induce the severe coverage of Pt surface by TiO$_x$, which may be due to pre-nucleation reduction method for the synthesis of the parent Pt/TiO₂[31,32].

The reaction profiles for BA hydrogenation display that reaction rate is faster with Pt/TiO₂ than with Pt/TiO₂-450 (Supplementary Fig. 6). Under similar conditions, Pt/TiO₂ with >99% conversion is more active than Pt/TiO₂-200 (59% conversion) and Pt/TiO₂-450 (10% conversion) (Table 1). To make reasonable comparisons, the TOFs of Pt/TiO₂ catalysts were normalized to Pt dispersion obtained with CO chemisorption. Pt/

TiO₂, Pt/TiO₂-200 and Pt/TiO₂-450, respectively, afford TOFs of 2200, 757, and 103 h⁻¹, further confirming that Pt/TiO₂ is more active than Pt/TiO₂-200 and Pt/TiO₂-450.

Generally, Pt/TiO₂ treated at high temperature under $H_2$ atmosphere would induce the change in electronic and geometric structure of Pt due to the strong metal-support interaction (SMSI) effect. In order to understand the different catalytic properties of Pt/TiO₂ catalysts, the electronic structure of Pt was first characterized by X-ray photoelectron spectroscopy (XPS) (Fig. 2c, Table 2). In comparison with Pt/TiO₂, Pt 4f binding energies (BEs) of Pt/TiO₂-200 and Pt/TiO₂-450 show an obvious downward shift, respectively, by 0.3 and 0.4 eV, suggesting that Pt/TiO₂ has more electron-deficient Pt site. The decrease in Pt 4f BEs indicates the charge transfer from Ti cations to Pt NPs induced by SMSI[33], which was further confirmed by the higher Ti $2p_{3/2}$ BEs of Pt/TiO₂ than those of Pt/TiO₂-200 and Pt/TiO₂-450 (Supplementary Fig. 7). It should be noted that Ti $2p_{3/2}$ BEs of Pt/TiO₂ are lower than those of TiO₂, implying the electron transfer from Pt to Ti cations. The $Pt^0/Pt^{\delta+}$ ratio of Pt/TiO₂ was increased from 68/32 to 74/26 after heat treatment in $H_2$, showing that the reduction degree of Pt increases at high temperature (Table 2). The electronic structure of Pt/TiO₂ catalysts could be facilely modified due to the electron-withdrawing ability of acidic TiO₂ support[34] and the SMSI effect of Pt–TiO₂ system[35].

The electronic and geometric structures of Pt NPs were further characterized with in situ diffuse reflectance infrared Fourier transform spectra (DRIFTS) of CO chemisorption (Fig. 2d). DRIFTS of adsorbed CO for Pt/TiO₂ shows four distinct $\upsilon_{CO}$

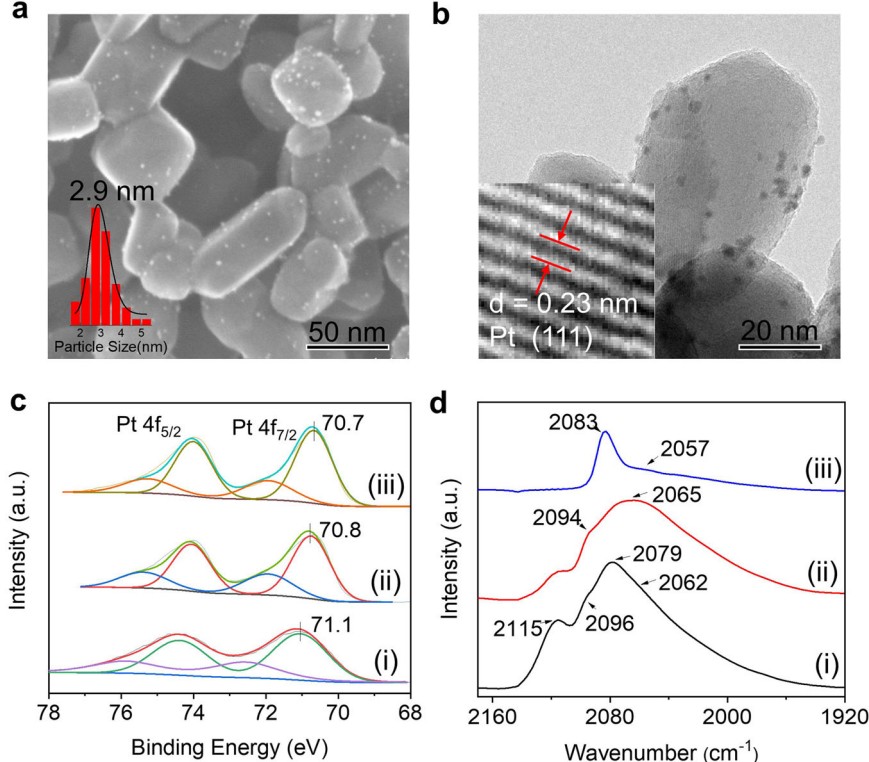

**Fig. 2 Characterization of Pt/TiO₂ catalysts. a** HRSEM image and **b** HRTEM image of Pt/TiO₂. **c** Pt 4f XPS core level spectra and **d** CO DRIFTS results for (**i**) Pt/TiO₂, (**ii**) Pt/TiO₂-200, and (**iii**) Pt/TiO₂-450.

**Table 2 Chemisorption and XPS results of Pt catalysts.**

| Cat. | Pt dispersion (%)[a] | Pt dispersion (%)[b] | Ti 2p$_{3/2}$ (eV)[c] | Pt 4f$_{7/2}$ (eV)[c] | Pt$^0$/Pt$^+$ (%)[c] |
|---|---|---|---|---|---|
| Pt/TiO₂ | 29.0 | 48.4 | 458.4 | 71.1 | 68/32 |
| Pt/TiO₂-200 | 22.6 | 49.6 | 458.6 | 70.8 | 70/30 |
| Pt/TiO₂-450 | 24.2 | 44.6 | 458.6 | 70.7 | 74/26 |
| TiO₂ | – | – | 458.5 | – | – |

[a]CO chemisorption results.
[b]H₂-O₂ titration results.
[c]Data obtained from XPS results.

bands in linear carbonyl region located at approximately 2115, 2096, 2079, and 2062 cm⁻¹. The band at 2115 cm⁻¹ can be assigned to Pt$^{\delta+}$[36]. The lower frequency vibrational stretch at 2062 cm⁻¹ is assigned to CO molecules adsorbed at low-coordination Pt-edge and -corner sites. The higher frequency vibrational stretch at 2096 and 2079 cm⁻¹ can be assigned to CO molecules that are adsorbed at the Pt (111) terrace sites (the coordination number of 9) and Pt (110) (the coordination number of 8), respectively[37–42]. The DRIFTS of adsorbed CO for Pt/TiO₂-200 is similar with that of Pt/TiO₂ with the exception that the red shift of the vibration peaks was observed, showing electron donation from Ti cations to Pt due to SMSI effect. The lower activity of Pt/TiO₂-200 than Pt/TiO₂ suggests that Pt with electron-deficient surface is favorable for the BA hydrogenation considering that the two catalysts have similar geometric surface structure of Pt.

The DRIFTS of adsorbed CO for Pt/TiO₂-450 is quite different from those of Pt/TiO₂ and Pt/TiO₂-200. The obvious change in peak intensities of Pt/TiO₂-450 suggests the reconstruction of surface Pt atoms under H₂ treatment at high temperature[39]. The relatively high peak intensity at 2083 cm⁻¹ indicates that Pt surface has more well-ordered Pt (111). The peak assigned to Pt

(111) is gradually red-shifted with H₂ treatment temperature increasing, implying the TiO₂ donates more electrons to Pt at higher temperature[43]. In comparison with Pt/TiO₂ and Pt/TiO₂-200, the much lower activity of Pt/TiO₂-450 indicates that the electron-deficient and low-coordination Pt sites may be active for BA hydrogenation.

The reaction orders of BA and H₂ were investigated with Pt/TiO₂ and Pt/TiO₂-450 as representative catalysts considering that the reaction kinetics are particularly sensitive to the Pt structure (Fig. 3a, b)[44,45]. To ensure the collection of reliable kinetic data, the system was verified to be free of mass transfer resistances (see Supporting Information 1). The order of BA hydrogenation with respect to BA is, respectively, −0.34 and +0.45 for Pt/TiO₂ and Pt/TiO₂-450, implying the stronger adsorption of BA on Pt/TiO₂ than on Pt/TiO₂-450. The reaction rate of Pt/TiO₂ is increased along with H₂ pressure and no plateau was observed with H₂ pressure from 4 to 20 bar, which is possibly related with the strong adsorption of BA. The reaction order with respect to H₂ for Pt/TiO₂ and Pt/TiO₂-450 is +0.54 and ~0, respectively. The positive order in hydrogen for the BA hydrogenation is a logical consequence of hydrogenation being involved in rate-determining step (RDS). The kinetic results show that the overall

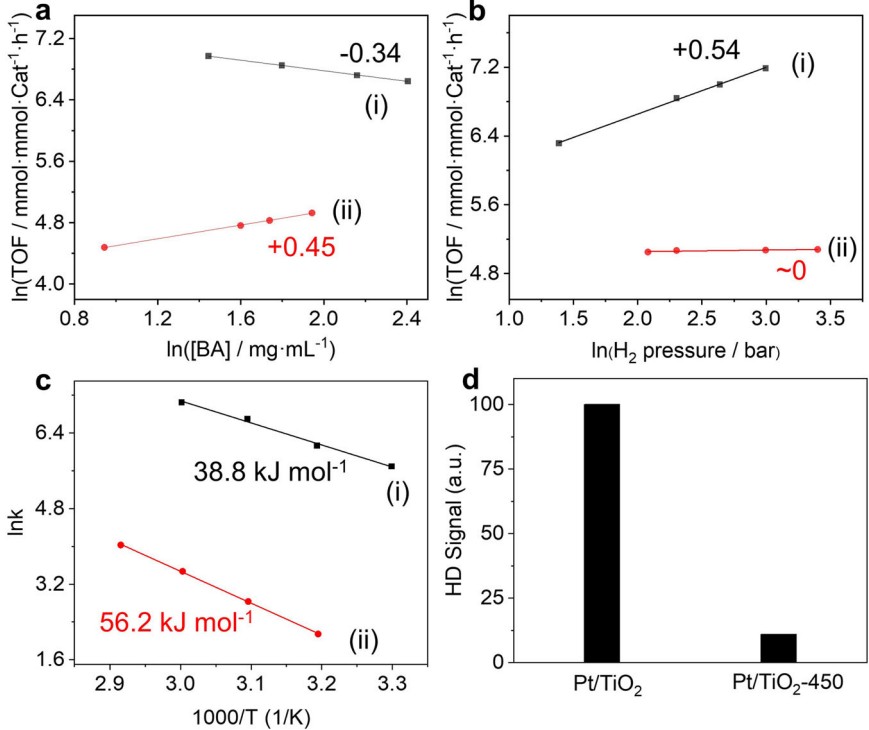

**Fig. 3 Kinetic results and H$_2$-D$_2$ exchange results on Pt/TiO$_2$ and Pt/TiO$_2$-450.** Reaction orders with respect to **a** BA and **b** H$_2$, **c** Arrhenius plots showing apparent activation barriers, and **d** H$_2$-D$_2$ exchange results of (**i**) Pt/TiO$_2$ and (**ii**) Pt/TiO$_2$-450. Reaction conditions for (**a**) and (**b**): 60 °C, 3 mL n-hexane, H$_2$ pressure: 1–30 bar, BA concentration: ~3–10 mg mL$^{-1}$. Reaction conditions for (**c**): $T$ = 30–70 °C, S/C = 250, 3 mL hexane. The BA conversion was maintained ~10–20% by adjusting reaction time. TOF turnover frequency, BA benzoic acid.

reaction order of BA hydrogenation on Pt/TiO$_2$-450 is much larger than that on Pt/TiO$_2$ (+0.47 vs. +0.20, Supplementary Table 4), indicating different reaction mechanisms for the two catalysts. Temperature-dependent reactivity measurements were taken to obtain apparent activation barriers with Pt/TiO$_2$ and Pt/TiO$_2$-450 as representatives (Fig. 3c). The activation energies for Pt/TiO$_2$ and Pt/TiO$_2$-450 are, respectively, ~38 and ~56 kJ mol$^{-1}$, showing the two catalysts have different active sites for BA hydrogenation[46–48]. The higher energy barriers of Pt/TiO$_2$-450 explains its low activity in BA hydrogenation.

The characterization data show that the electron density of Pt NPs follows the order of Pt/TiO$_2$ < Pt/TiO$_2$-200 < Pt/TiO$_2$-450. In combination with the catalytic activity, we can infer that the electronic deficient Pt may favor the high BA hydrogenation. From kinetic data, BA is strongly adsorbed on Pt/TiO$_2$ with electron-rich Pt sites. According to the Sabatier rule, too strong or too weak adsorption of reactants on catalysts both disfavor high activity. The minus reaction order of BA over Pt/TiO$_2$ means that BA (or CCA) in fact is "a poison" for the catalyst, which blocked the active sites[49]. The higher reaction order of H$_2$ over Pt/TiO$_2$ than over Pt/TiO$_2$-450 indicated that hydrogen adsorption is relative more difficult on the former sample. But Pt/TiO$_2$ still shows much higher activity than Pt/TiO$_2$-450, suggesting that the hydrogen activation plays an important role in BA hydrogenation. The H$_2$ activation ability of the catalysts was measured using H$_2$-D$_2$ exchange experiments (Fig. 3d, Supplementary Table 5). The normalized HD formation rate of Pt/TiO$_2$ is more than 10-fold that of Pt/TiO$_2$-450, showing that Pt/TiO$_2$ with electronic deficient Pt surface is more active for H$_2$ activation than Pt/TiO$_2$-450 with electronic rich Pt surface. This result suggests that the high activity of Pt/TiO$_2$ is partly attributed to the high H$_2$ dissociation capacity.

Pt/TiO$_2$ treated at high temperature under reductive atmosphere would induce a significant change in surface structure of

TiO$_2$[30], which may influence the H$_2$ spillover[50]. H$_2$-TPD measurement was performed to reveal the hydrogen species formed on Pt/TiO$_2$ and Pt/TiO$_2$-450 (Supplementary Fig. 8). For Pt/TiO$_2$, the low-temperature peak can be assigned to the desorption of H$_2$ on metallic Pt and the peaks above 200 °C are related to hydrogen species on TiO$_2$ derived from hydrogen spillover[30,51]. However, only two broad and weak peaks were observed for Pt/TiO$_2$-450, showing the difficulty in H$_2$ spillover possibly due to the dehydroxylation of TiO$_2$ during heat treatment[30]. The facile H$_2$ spillover on Pt/TiO$_2$ may also contribute to the high activity.

**The role of the carboxyl group.** Generally, the deficient phenyl ring does not easily bind on metal surface[20], which always results in low catalytic activity[52]. However, the high activity of Pt/TiO$_2$ in BA hydrogenation suggests that the carboxyl group may affect the hydrogenation activity. To identify the role of carboxyl group in BA hydrogenation, hydrogenation of benzotrifluoride and toluene were conducted (Fig. 4a). Pt/TiO$_2$ could efficiently catalyze the hydrogenation of the above substrates to the corresponding aromatic ring hydrogenated products. The activity followed the order of BA > toluene > benzotrifluoride under similar reaction conditions. The unusual high activity of BA hydrogenation is in contrast to the previous findings that the deficient phenyl ring was difficult to be hydrogenated. This suggests that the carboxyl group may be involved in the whole reaction process although it cannot easily be hydrogenated at mild reaction conditions[10].

DFT calculation results show that the benzene ring adsorption in parallel to the metal plane is the most favorable adsorption configuration of toluene on Pt (111), and the methyl group is far away from the Pt surface. This is mainly due to better superposition of its π-orbitals with the Pt conduction band (Fig. 4b)[53]. Different from toluene, BA molecule adopted a

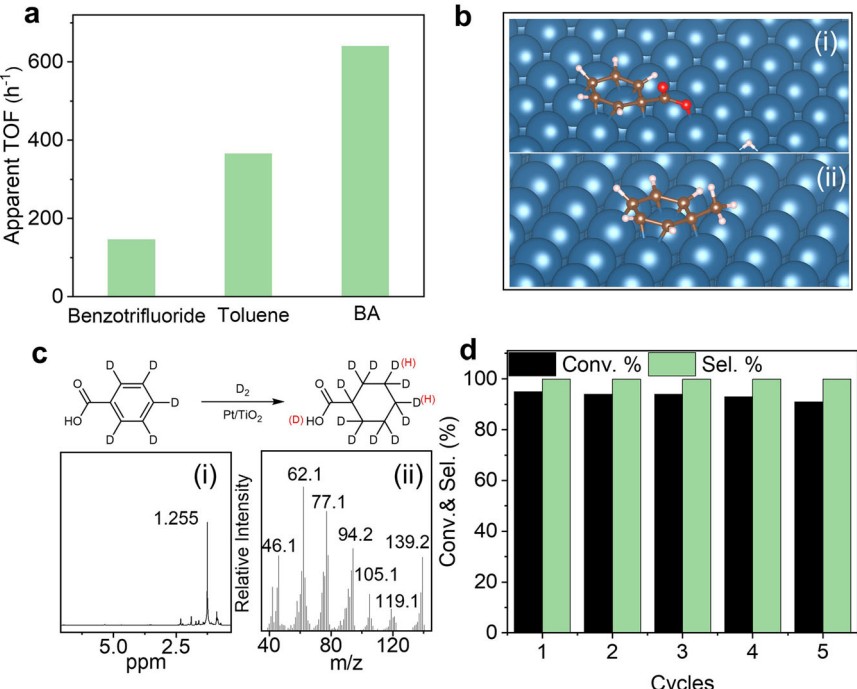

**Fig. 4 Catalytic mechanism and stability test. a** Comparison of the catalytic activity of Pt/TiO₂ in hydrogenation of benzotrifluoride, toluene, and BA. **b** The adsorption mode of (**i**) BA and (**ii**) toluene on Pt (111). **c** ¹H-NMR (**i**) and MS analysis (**ii**) of the product for benzoic-d[5] acid hydrogenation with D₂. **d** Recycling stability of Pt/TiO₂ in the hydrogenation of BA. BA benzoic acid, MS mass spectrum.

configuration that an O atom of the carboxyl group is coadsorbed on the Pt surface. The strong adsorption of BA on Pt/TiO₂ as discussed above may be derived from the coadsorption of carboxyl group on Pt surface[54]. The dissociated H from the carboxyl group may act as one of the H sources. To confirm this, a control experiment was conducted by using benzoic acid-d[5] and D₂ as reactants. The mass spectrum (MS) analysis of the product shows the appearance of molecular ion peaks at *m/z* of 139.2 and 138.2 with the intensity ratio of 1.9, denoting the presence of 6 deuterated and 5 deuterated CCA in the product (Fig. 4c). An obvious sharp peak at 1.255 was observed in the ¹H-NMR spectrum of the product assigned to the H on the m- or p-position of cyclohexane ring (Fig. 4c), further confirming the results of MS analysis. The above results show that the dissociated H from carboxyl group is involved in the hydrogenation process. To this end, the active Pt–H species from homolytic dissociation of hydrogen and dissociated H from carboxyl group attack the activated BA molecule to produce CCA. Besides, the adsorption of carboxyl group on Pt/TiO₂ favors the orientation of the aromatic ring on the Pt surface, which may facilitate the hydrogen transfer from Pt surface to BA molecules[55]. On the basis of this mechanism, the Pt with electron-deficient surface favors the adsorption and dissociation of carboxyl group, which could enhance BA hydrogenation activity.

The stability of Pt/TiO₂ was tested in BA hydrogenation. During 5 cycles, no obvious decrease in conversion and selectivity could be observed (Fig. 4d). To identify the structure and composition of the used catalysts, Pt/TiO₂ after 5 cycles was characterized by TEM and XPS techniques. The results show that used Pt/TiO₂ has similar particle size and electronic structure as the fresh one (Supplementary Fig. 9), implying no obvious change in structure and composition during recycling process. The hot filtration reaction was performed by removing Pt/TiO₂ from the reaction suspension after BA conversion reaching 50%. Then the filtrate was recharged with 10 bar H₂. After 48 min, no further increase in conversion was observed (Supplementary Fig. 6).

Meanwhile, the concentrations of Pt in the reaction solution were below the detection limit of ICP-AES, confirming the heterogeneous nature of Pt/TiO₂. The above results confirm the high stability of Pt/TiO₂ during recycling process.

**Substrate scopes.** Pt/TiO₂ was also used for hydrogenation of BA derivatives at mild conditions (Table 3). First, the hydrogenations of methyl-substituted BA (o-, m-, and p-), p-ethyl benzoic acid, and p-pentyl benzoic acid were investigated and the full conversion was obtained in 6 h with the kinetically favored *cis*-isomer[56]. The *cis/trans*-ratio varied in the range of 20/30 to 86/14 (Table 3, entries 1–5, Supplementary Figs. 10–14). For p-isopropyl benzoic acid, it needs 10 h to reach full conversion with the *cis/trans*-ratio of 68/32 (Table 3, entry 6, Supplementary Fig. 15). Even for p-trifluoromethylbenzoic acid with more electron-deficient aromatic ring, full conversion was achieved in 10 h though S/C ratio was decreased to 50/1 (Table 3, entry 7, Supplementary Fig. 16), demonstrating the high activity of Pt/TiO₂. p-Hydroxylbenzoic acid, phenyl propionic acid, and phenyl pentanoic acid could be efficiently transferred to corresponding products over Pt/TiO₂ (Table 3, entries 8–10, Supplementary Figs. 17–19). The hydrogenation of methyl benzoate, monomethyl terephthalate, and dioctyl phthalate resulted in the formation of the aromatic hydrogenated products using Pt/TiO₂ as catalyst (Table 3, entries 11–13, Supplementary Figs. 20–22). It should be mentioned that Pt/TiO₂ could also catalyze the hydrogenation of terephthalic acid, phthalic acid, isophthalic acid, and even the challenging trimesic acid and trimethyl trimesate to corresponding aromatic ring saturated product under mild conditions, further demonstrating the high efficiency of Pt/TiO₂ for the hydrogenation of aromatic acids (Table 3, entries 14–18, Supplementary Figs. 23–27). The hydrogenation of dioctyl phthalate/phthalate acid and trimesic acid/trimethyl trimesate, respectively, produces the *trans*- and *cis*-isomers, and the hydrogenation of the other substrates investigated in this paper results in the formation of *cis*-isomer as the main product on the

**Table 3 Hydrogenation of BA derivatives using Pt/TiO$_2$ as catalyst[a].**

| Entry | Substrate | Product | Time (h) | Conv. (%) | Sel. (%)[b] |
|---|---|---|---|---|---|
| 1 | (4-methylbenzoic acid) | (4-methylcyclohexanecarboxylic acid) | 6 | > 99 | 99 (70:30) |
| 2 | (3-methylbenzoic acid) | (3-methylcyclohexanecarboxylic acid) | 6 | > 99 | 99 (79:21) |
| 3 | (2-methylbenzoic acid) | (2-methylcyclohexanecarboxylic acid) | 6 | > 99 | 99 (86:14) |
| 4 | (4-ethylbenzoic acid) | (4-ethylcyclohexanecarboxylic acid) | 6 | > 99 | 99 (72:28) |
| 5 | (4-propylbenzoic acid) | (4-propylcyclohexanecarboxylic acid) | 6 | > 99 | 99 (72:28) |
| 6 | (4-isopropylbenzoic acid) | (4-isopropylcyclohexanecarboxylic acid) | 10 | > 99 | 99 (68:32) |
| 7[d] | F$_3$C— COOH | F$_3$C— COOH | 10 | > 99 | > 99 (64:36) |
| 8[e] | HO— COOH | HO— COOH | 12 | > 99 | > 99 (65:35)[c] |
| 9 | (phenylacetic acid deriv.) COOH | COOH | 6 | > 99 | > 99 |
| 10[d] | COOH | COOH | 6 | > 99 | > 99 |
| 11 | (methyl benzoate) | | 6 | > 99 | > 99 |
| 12[f] | MeOOC— COOH | MeOOC— COOH | 10 | > 99 | 99% (66:34) |
| 13[g] | C$_8$H$_{17}$OOC  COOC$_8$H$_{17}$ | C$_8$H$_{17}$OOC  COOC$_8$H$_{17}$ | 3 | > 99 | 99% (0:~100) |
| 14[h] | HOOC  COOH | HOOC  COOH | 2 | > 99 | 99% (0:~100) |
| 15[i] | HOOC— COOH | HOOC— COOH | 3 | > 99 | 99% (77:23) |
| 16[j] | HOOC— COOH | HOOC— COOH | 3 | > 99 | 99% (77:23) |
| 17[i] | COOH / HOOC  COOH | COOH / HOOC  COOH | 3 | > 99 | 99% (~100:0) |
| 18[g] | COOMe / MeOOC  COOMe | COOMe / MeOOC  COOMe | 3 | > 99 | 99% (~100:0) |

[a]Reaction conditions: 40 °C, 10 bar H$_2$, S/C = 250, 3 mL hexane.
[b]Selectivity to ring hydrogenation product; the *cis/trans*-ratio in the parentheses was determined by $^1$H-NMR results (Supplementary Figs. 10–27)[66].
[c]Chair conformation.
[d]S/C = 50.
[e]3-mL H$_2$O.
[f]1.5 mL hexane and 1.5 mL acetic acid.
[g]60 °C.
[h]60 °C, 1.5 mL H$_2$O, and 1.5 mL acetic acid.
[i]60 °C, 20 bar H$_2$, 3 mL solvent (10 v/v% H$_2$O in n-hexane).
[j]80 °C, 20 bar H$_2$, 3 mL hexane (10 v/v% H$_2$O in n-hexane).

basis of NMR analysis, which may be caused by the steric hindrance effect[4].

In conclusion, Pt/TiO$_2$ was found to be a superior catalyst for BA hydrogenation in comparison with Ru/C and Pd/C due to the weak interaction strength between Pt and BA which inhibits the toxic of BA to the catalyst. A record TOF of 4490 h$^{-1}$ was achieved with Pt/TiO$_2$ under 80 °C and 50 bar H$_2$ in hexane, more than 10 times higher than the literature results under similar conditions. Isotopic studies confirm that the dissociated H

from the carboxyl group is involved in BA hydrogenation which could be facilitated by the strong adsorption of BA on Pt surface. By comparing the activity of Pt/TiO$_2$ catalysts with different surface electronic and geometric structures, it could be concluded that electron-deficient and low-coordination Pt sites show higher activity than electron-rich and high coordination Pt sites in BA hydrogenation, possibly due to the combined effect of higher H$_2$ activation ability and the stronger adsorption of BA at electron-deficient Pt sites. The wide substrate scope including very

challenging terephthalic acid, phthalate acid, phthalic acid, isophthalic, and trimesic acid demonstrates the potential practical applications of Pt/TiO$_2$ in hydrogenation of BA and its derivatives.

## Methods

**Preparation of the Pt/TiO$_2$.** Pt catalysts with Pt loading of 2 wt% were prepared by the deposition precipitation method using NaBH$_4$ as the reductant[34,57,58]. Typically, 200 mg of TiO$_2$ and the desired amount of H$_2$PtCl$_6$ (4 mg Pt) was initially dispersed into 50 mL of aqueous solution. After stirring for 1 h at room temperature, a freshly prepared NaBH$_4$ aqueous solution (2.3 mg, 0.2 mg mL$^{-1}$) was added slowly. After stirring for another 1 h, the solid was collected by filtration and washed with water and ethanol three times. Finally, the obtained powder was dried at room temperature overnight. The catalyst was denoted as Pt/TiO$_2$.

**Preparation of the Pt/TiO$_2$-200 and Pt/TiO$_2$-450.** Pt/TiO$_2$ was treated in H$_2$ atmosphere with a flow rate of 20 mL min$^{-1}$ at the desired temperature for 2 h with a heating rate of 1 °C min$^{-1}$. The sample after treatment was denoted as Pt/TiO$_2$-$T$, where $T$ (200 and 450) refers to the treatment temperature.

**Synthesis of other oxide support loaded Pt catalyst.** Pt/SiO$_2$ with Pt loading of 2 wt% was prepared by the wet impregnation method by dispersing SiO$_2$ (200 mg) in 2 mL of aqueous solution of H$_2$PtCl$_6$ (4 mg Pt) for 5 h. Then the solid product after drying by an evaporator and reduced under H$_2$ atmosphere at 300 °C for 2 h to afford Pt/SiO$_2$. Other oxide-supported Pt catalysts were prepared with a similar method to Pt/TiO$_2$ except that the corresponding oxide was used as supports. Analysis by inductively coupled plasma atomic emission analysis (ICP-AES) clearly indicated that the desired amounts of metal species were successfully loaded onto each of the catalysts.

**Catalyst characterization.** Transmission electron microscopy (TEM) image were obtained using a HITACHI HT7700 at an acceleration voltage of 100 kV. High-resolution scanning electron microscopy (HRSEM) was undertaken by using a HITACHI S5500 apparatus operating at an acceleration voltage of 1–30 kV. X-ray photoelectron (XPS) was performed on an ESCALAB 250xi spectrometer using Al K$_\alpha$ radiation. All the XPS spectra were calibrated by the C1s peak (284.6 eV) from contamination to compensate the charge effects. N$_2$ sorption isotherms were carried out on a Micromeritics ASAP2020 volumetric adsorption analyzer. Liquid $^1$H and $^{13}$C, NMR spectra were recorded on a Bruker Avance 400 MHz spectrometer at 25 °C.

**In situ DRIFTS.** In situ diffuse reflectance infrared Fourier transform spectra (DRIFTS) of CO chemisorption was measured on a Thermo Scientific IR spectrometer with a mercury cadmium telluride (MCT) detector, recorded with a resolution of 4 cm$^{-1}$ [59]. Prior to CO adsorption, the samples were treated in situ in the DRIFT cell under H$_2$ flow (20 mL min$^{-1}$) at the desired temperatures for 1 h, followed by purging with a He flow at the same temperature for 30 min. After cooling to room temperature, a background spectrum was collected. Then the He flow was switched to a pure CO flow (20 mL min$^{-1}$) until saturated adsorption was achieved. CO-adsorption experiments were carried out sequentially on a single sample. Gas-phase CO spectra were collected at the same pressure and subtracted from the corresponding sample spectra.

**CO and H$_2$ chemisorption experiments.** CO/H$_2$ chemisorption measurement was performed at 50 °C on Autochem II 2920 chemisorption instrument with a thermal conductivity detector (TCD). For CO chemisorption, the sample (~100 mg) was pretreated with hydrogen at desired temperatures for 1 h, followed by purging with high-purity He for 30 min. After the sample was cooled down to 50 °C, a 5% CO/He mixture was injected into the reactor repeatedly until CO adsorption was saturated. The dispersion of Pt was calculated from the amount of CO adsorbed by assuming the CO/Pt adsorption stoichiometry to be 1/1. Pt dispersion obtained from H$_2$ chemisorption was measured by a HOT method (H$_2$-O$_2$ titration)[60]. Typically, 100 mg of Pt catalyst was reduced in 5 vol% H$_2$/Ar at 120, 200, and 450 °C for 2 h, respectively, for Pt/TiO$_2$, Pt/TiO$_2$-200, and Pt/TiO$_2$-450. The sample was then cooled to the 140 °C under a flow rate of 60 mL/min of Ar. Then plus O$_2$ was introduced into the carrier gas until O$_2$ peak reached saturation to completely oxide the surface Pt and followed by H$_2$ reduction. Assuming that one hydrogen molecule reduced one surface PtO to Pt and 0.5 hydrogen adsorbed on one Pt atom, the Pt dispersion was calculated as follows:

$$D_{Pt} = \frac{N_{surface}}{N_{total}} = \frac{\frac{2}{3}\ \text{amount of H}_2}{N_{total}} \times 100$$

**H$_2$-TPD experiments.** H$_2$-TPD experiments were conducted in a U-type quartz tube connected to a mass spectrometer (Autochem 2910). Hundred milligrams of catalyst sample was placed in a U-type quartz tube, heated to 120 and 450 °C, respectively, for Pt/TiO$_2$ and Pt/TiO$_2$-450 and kept for 60 min in Ar flow (30 mL

min$^{-1}$) to remove adsorbed species from catalyst surface. When the sample was cooled down to 25 °C, the flow was switched to H$_2$ (30 ml min$^{-1}$) for 60 min, followed by purging with Ar (30 ml min$^{-1}$) for 40 min. The sample was then heated to 800 °C with a ramp rate of 10 °C min$^{-1}$ in Ar flow and the TPD profiles were recorded simultaneously.

**H$_2$−D$_2$ exchange.** H$_2$−D$_2$ exchange reactions were carried out in a flow quartz reactor at 22 °C[61]. The formation rate of HD was measured by mass signal intensity (ion current). Before the test, the catalysts were heated in H$_2$ (10 mL min$^{-1}$) at 200 °C for 20 min. After the sample was cooled down to room temperature, D$_2$ (10 mL min$^{-1}$) mixed with H$_2$ was passed through the sample. The gas hourly space velocity (GHSV) is 2.9 × 10$^7$ mL h$^{-1}$ g$_{metal}$$^{-1}$. Under these conditions, the H$_2$-D$_2$ exchange conversions were always kept below 10% for calculation of TOF. Products (HD, H$_2$, and D$_2$) were analyzed with an online mass spectrometer (GAM200, InProcess Instruments). The mass/charge ratio ($m/z$) values used are 2 for H$_2$, 4 for D$_2$, and 3 for HD. The background HD exchanges from the corresponding support were deducted from the results.

**Hydrogenation test.** The hydrogenation reactions were carried out in a stainless steel autoclave (300 mL) with a thermocouple-probed detector. In a typical process for benzoic acid (BA) hydrogenation, a desired amount of the solid catalyst was placed in an ampule tube, followed by the addition of BA (0.12 mmol) and 3 mL of n-hexane (for reaction performed at S/C of 1200, 0.3 mL of acetic acid was added to assist the dissolution of BA). The ampule tube was loaded into the reactor. After the tube was purged six times with hydrogen, the final pressure was adjusted to 10 bar and the reactor was heated to the desired temperature with vigorous stirring. After the reaction, the solid catalyst was separated by centrifugation and the filtrate was collected, diluted with n-hexane, and analyzed by an Agilent 6890N GC instrument equipped with an Agilent J&W GC HP-INNOWax capillary column (30 m × 0.32 mm × 0.25 μm). The diphenyl ether was used as the internal standard to determine the conversion, selectivity, and carbon balance. The carbon balance was ~100%. For the recycle experiments, the liquid was decanted after centrifugation of the reaction mixture. The residual catalyst was thoroughly washed with n-hexane, and used directly for the next run.

**Computational setup.** All the calculations were performed with density functional theory (DFT) by using the Vienna Ab-initio Simulation Package (VASP)[62,63]. The projector augmented-wave pseudopotential method with Perdew–Burke–Ernzerhof (PBE) exchange-correlation functional including zero-damping DFT-D3 of Grimme's correction was employed[64,65]. A plane-wave basis with cutoff energy of 400 eV was adopted. Four-atomic-layer slab models of Pt(111), Pd(111), and Ru (0001) with the bottom two layers fixed consisting of 144 metal atoms were built. The vacuum spaces were set as 15 Å between the layers. A gamma k-point sampling of 1 × 1 × 1 was selected. The convergence energy and force were set to be 1 × 10$^{-5}$ eV and 0.02 eV/Å, respectively. The optimized fractional coordinates for different adsorbates on Pt(111)/Pd(111)/Ru(1000) surface see Supplementary Data 1.

The adsorption energy $E_{ad}$ was calculated as:

$$E_{ad} = E_{ad/sub} - E_{mol} - E_{sub}$$

The $E_{ads/sub}$, $E_{mol}$, and $E_{sub}$ are the total energy of the adsorbed molecule, the molecule in the gas phase, and the pure slab, respectively.

## Data availability
Any relevant data are available from the authors upon reasonable request.

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

## Acknowledgements

We acknowledge financial support from the National Key R&D Program of China (2017YFB0702800), the National Natural Science Foundation of China (21733009), and the Strategic Priority Research Program of the Chinese Academy of Sciences (XDB17020200).

## Author contributions

Q.H.Y. conceived the idea. Q.H.Y., M.G., X.T.K., and C.Z.L. co-wrote the paper. M.G. synthesized the nanomaterials and carried out the catalysis experiments. X.T.K. carried out the model construction and DFT calculations. C.Z.L. analyzed part of NMR results. All the authors contributed to the overall scientific interpretation and edited the manuscript.

## Competing interests

The authors declare no competing interests.
