## [Peer Review File · Communications Chemistry]

Reviewers' comments:

Reviewer #1 (Remarks to the Author):

This manuscript reports the synthesis of Pt nanoparticles (ca. 2.9 nm) on TiO₂ supports for the catalytic hydrogenation of benzoic acid (BA) to cyclohexanecarboxylic acid (and also of BA derivatives). The Pt/TiO₂ catalyst shows superior catalytic performance with a record TOF of 4490 h⁻¹ under mild conditions, which comes from the synergistic effect of the stronger adsorption of BA and higher H₂ activation ability by electron deficient Pt sites. The research is well-executed, with the pertinent catalyst characterization, blank experiments, and catalyst, reaction condition and substrate scope. Computational studies support the key role of the effect of the electron deficient Pt sites on the catalytic performance. Although the design of the Pt/TiO₂ catalyst is not terrifically novel, I think this manuscript is suitable for publication in Communications Chemistry. Some issues to be addressed before publication:

- 1、 Why water may block the active sites of the Pt catalyst? Please explain and include in the manuscript.
- 2、 The Pd and Ru catalysts used to compare the catalytic performance are not comprehensive, it's necessary to supplement the performance test results of Pd/TiO₂ and Ru/TiO₂.
- 3、 The dispersion results of Pt calculated by H₂ and CO chemisorption are quite different. Please check and specify.
- 4、 The drawing and layout of the diagrams are not standardized in this article, such as Figure 2.
- 5、 The reaction temperature should be unified when calculating the reaction orders of the reactants.
- 6、 Do the structure and composition of the catalysts change during the reaction? So it's necessary to provide more characterization of the catalyst after the reaction.
- 7、 There are many spelling and grammar mistakes in the article, which need to be thoroughly corrected. The references also have the problem of repeated citations.

Reviewer #2 (Remarks to the Author):

This ms reports on a study on benzoic acid hydrogenation on Pt/TiO₂ catalysts. The reaction is difficult and the catalytic performances are very good compared to benchmark catalysts. I have read this contribution with great interest and have considered it for publication, but further evidences are still required to strengthen the conclusions.

1) Hydrogenation reactions in multi-phase reactors can often be mass transfer limited and I find the argument presented lines 220-222 unsatisfactory. There are well-established protocols for discerning the role of intra-phase and interphase mass transfer limitations in catalytic systems (e.g., Madon-Boudart, Weisz Prater, Mears, among others). If the authors can clearly discern that the effects shown (lower E_a) represent kinetic events involving bond forming/bond cleavage and not physical rate processes involving mass transfer then the drawn conclusion can be made rigorously.

2) I also find the explanation given on the greater ease in activating hydrogen on an electron-deficient metallic species surprising. This goes against what is generally accepted: see for example the extreme case of single atom catalysts very deficient in electrons, which are only able to activate H₂ via heterolytic activation involving the support. If the activation of the carboxylic function can contribute to increasing the rate of hydrogenation, I do not think that it is sufficient to explain the observed differences. Indeed, esters are also hydrogenated very quickly (Table 3). It would certainly be relevant to study the possibility of H-spillover (for example by H₂-TPD/MS) on the two catalysts Pt/TiO₂ and Pt/TiO₂-450. In fact, the reduction of rutile TiO₂ at 450 ° C must deeply modify the surface of the rutile support (oxygen vacancy formation etc...). H₂ spillover enhanced hydrogenation capability of Pt/TiO₂ catalyst could be an explanation of the observed

higher activity of the Pt/TiO₂ catalyst.

Reviewer #3 (Remarks to the Author):

The authors have nicely presented the work entitled "Pt/TiO₂ Catalyzed Hydrogenation of Benzoic Acid with Unprecedented High Activity" where they have claimed very high activity of Pt/TiO₂ for the conversion of Benzoic acid (BA), terephthalic acid, iso-phthalic acid, trimesic acid and other BA derivatives to corresponding aromatic saturated products. The whole work is very well designed and performed.

The present manuscript may be considered for publication in the Communications Chemistry journal after consideration of suggested revisions.

Some critical remarks are:

1. The author should relook the title, needs modification. They should avoid the terminology such as "Unprecedented High Activity" in the title. The author also mentioned "benzoic acid" in title but in substrates they have used some other aryl substituted alkyl acids, may consider in title and make it general.
2. The scale bar size (i.e. xyz nm) should also have to be written on HRSEM and HAADF-STEM images in Figure 2 of the manuscript and Figure S1- Figure S3 and Figure S6 of supporting information for better data communication to readers.
3. The Selective Area Electron Diffraction (SAED) and X-Ray Diffraction (XRD) characterization of Pt/TiO₂ NPs should also be included in the manuscript for the confirmation of active planes responsible for desired transformation.
4. The authors should also provide the hot filtration test and Inductively coupled plasma atomic emission spectroscopy (ICP-AES) analysis details of Pt/TiO₂ catalyst for clarification regarding true heterogeneous nature of the catalyst and Pt metal leaching during the course of reaction respectively.
5. In Page No. 18, Line No. 352, 'NaBH₄ as the reactant' should be replaced with 'NaBH₄ as the reductant'.
6. Authors have cited two research papers (Ref. 34, 58) published in 2018-2019 approaching Pt metal deposition precipitation method using NaBH₄ as the reductant (Page No. 18, Line No. 352) whereas, a work published by Das et al. using NaBH₄ as the reductant for solid supported Pt NPs preparation through deposition precipitation method in 2013 (Green Chem., 2013,15, 3421-3428) should be considered in the text and reference.
7. Although authors have very well explained the role of carboxylic group on phenyl ring for hydrogenation reaction. However, have the authors tried other functional groups such as aldehydes, ketones and alcohols for this protocol. If yes, then they should mention the results in one line.

Reviewers' comments:

Reviewer #1 (Remarks to the Author):

This manuscript reports the synthesis of Pt nanoparticles (ca. 2.9 nm) on TiO₂ supports for the catalytic hydrogenation of benzoic acid (BA) to cyclohexanecarboxylic acid (and also of BA derivatives). The Pt/TiO₂ catalyst shows superior catalytic performance with a record TOF of 4490 h⁻¹ under mild conditions, which comes from the synergistic effect of the stronger adsorption of BA and higher H₂ activation ability by electron deficient Pt sites. The research is well-executed, with the pertinent catalyst characterization, blank experiments, and catalyst, reaction condition and substrate scope. Computational studies support the key role of the effect of the electron deficient Pt sites on the catalytic performance. Although the design of the Pt/TiO₂ catalyst is not terrifically novel, I think this manuscript is suitable for publication in Communications Chemistry. Some issues to be addressed before publication:

- 1、 Why water may block the active sites of the Pt catalyst? Please explain and include in the manuscript.
- 2、 The Pd and Ru catalysts used to compare the catalytic performance are not comprehensive, it's necessary to supplement the performance test results of Pd/TiO₂ and Ru/TiO₂.
- 3、 The dispersion results of Pt calculated by H₂ and CO chemisorption are quite different. Please check and specify.
- 4、 The drawing and layout of the diagrams are not standardized in this article, such as Figure 2.
- 5、 The reaction temperature should be unified when calculating the reaction orders of the reactants.
- 6、 Do the structure and composition of the catalysts change during the reaction? So it's necessary to provide more characterization of the catalyst after the reaction.
- 7、 There are many spelling and grammar mistakes in the article, which need to be thoroughly corrected. The references also have the problem of repeated citations.

Reviewer #2 (Remarks to the Author):

This ms reports on a study on benzoic acid hydrogenation on Pt/TiO₂ catalysts. The reaction is difficult and the catalytic performances are very good compared to benchmark catalysts.

I have read this contribution with great interest and have considered it for publication, but further evidences are still required to strengthen the conclusions.

1) Hydrogenation reactions in multi-phase reactors can often be mass transfer limited and I find the argument presented lines 220-222 unsatisfactory. There are well-established protocols for discerning the role of intra-phase and interphase mass transfer limitations in catalytic systems (e.g., Madon-Boudart, Weisz Prater, Mears, among others). If the authors can clearly discern that the effects shown (lower E_a) represent kinetic events involving bond forming/bond cleavage and not physical rate processes involving mass transfer then the drawn conclusion can be made rigorously.

2) I also find the explanation given on the greater ease in activating hydrogen on an electron-deficient metallic species surprising. This goes against what is generally accepted: see for example the extreme case of single atom catalysts very deficient in electrons, which are only able to activate H_2 via heterolytic activation involving the support. If the activation of the carboxylic function can contribute to increasing the rate of hydrogenation, I do not think that it is sufficient to explain the observed differences. Indeed, esters are also hydrogenated very quickly (Table 3). It would certainly be relevant to study the possibility of H-spillover (for example by H_2 -TPD/MS) on the two catalysts Pt/TiO₂ and Pt/TiO₂-450. In fact, the reduction of rutile TiO₂ at 450 ° C must deeply modify the surface of the rutile support (oxygen vacancy formation etc...). H_2 spillover enhanced hydrogenation capability of Pt/TiO₂ catalyst could be an explanation of the observed higher activity of the Pt/TiO₂ catalyst.

Reviewer #3 (Remarks to the Author):

The authors have nicely presented the work entitled “Pt/TiO₂ Catalyzed Hydrogenation of Benzoic Acid with Unprecedented High Activity” where they have claimed very high activity of Pt/TiO₂ for the conversion of Benzoic acid (BA), terephthalic acid, iso-phthalic acid, trimesic acid and other BA derivatives to corresponding aromatic saturated products. The whole work is very well designed and performed.

The present manuscript may be considered for publication in the Communications Chemistry journal after consideration of suggested revisions.

Some critical remarks are:

1. The author should relook the title, needs modification. They should avoid the terminology such as “Unprecedented High Activity” in the title. The author also mentioned “benzoic acid” in title but in substrates they have used some other aryl substituted alkyl acids, may consider in title and make it general.
2. The scale bar size (i.e. xyz nm) should also have to be written on HRSEM and HAADF-STEM images in Figure 2 of the manuscript and Figure S1- Figure S3 and Figure S6 of supporting information for better data communication to readers.

3. The Selective Area Electron Diffraction (SAED) and X-Ray Diffraction (XRD) characterization of Pt/TiO₂ NPs should also be included in the manuscript for the confirmation of active planes responsible for desired transformation.
4. The authors should also provide the hot filtration test and Inductively coupled plasma atomic emission spectroscopy (ICP-AES) analysis details of Pt/TiO₂ catalyst for clarification regarding true heterogeneous nature of the catalyst and Pt metal leaching during the course of reaction respectively.
5. In Page No. 18, Line No. 352, 'NaBH₄ as the reactant' should be replaced with 'NaBH₄ as the reductant'.
6. Authors have cited two research papers (Ref. 34, 58) published in 2018-2019 approaching Pt metal deposition precipitation method using NaBH₄ as the reductant (Page No. 18, Line No. 352) whereas, a work published by Das et al. using NaBH₄ as the reductant for solid supported Pt NPs preparation through deposition precipitation method in 2013 (Green Chem., 2013,15, 3421-3428) should be considered in the text and reference.
7. Although authors have very well explained the role of carboxylic group on phenyl ring for hydrogenation reaction. However, have the authors tried other functional groups such as aldehydes, ketones and alcohols for this protocol. If yes, then they should mention the results in one line.

Response to the Reviewers' comments

Responses to Reviewer #1

Comments:

This manuscript reports the synthesis of Pt nanoparticles (ca. 2.9 nm) on TiO₂ supports for the catalytic hydrogenation of benzoic acid (BA) to cyclohexanecarboxylic acid (and also of BA derivatives). The Pt/TiO₂ catalyst shows superior catalytic performance with a record TOF of 4490 h⁻¹ under mild conditions, which comes from the synergistic effect of the stronger adsorption of BA and higher H₂ activation ability by electron deficient Pt sites. The research is well-executed, with the pertinent catalyst characterization, blank experiments, and catalyst, reaction condition and substrate scope. Computational studies support the key role of the effect of the electron deficient Pt sites on the catalytic performance. Although the design of the Pt/TiO₂ catalyst is not terrifically novel, I think this manuscript is suitable for publication in Communications Chemistry. Some issues to be addressed before publication:

Response: Thank you so much for your careful review and valuable suggestions. The questions are answered point-by-point as follows:

Question 1: “Why water may block the active sites of the Pt catalyst? Please explain and include in the manuscript.”

Response: The H₂O (or dissociated OH species) could be strongly adsorbed on Pt surface, which may block the Pt surface active sites¹⁶. This sentence has been added in the revised manuscript.

Question 2: “The Pd and Ru catalysts used to compare the catalytic performance are not comprehensive, it’s necessary to supplement the performance test results of Pd/TiO₂ and Ru/TiO₂.”

Response: Thanks for your good suggest. Pd/TiO₂ and Ru/TiO₂ were prepared using the same method with Pt/TiO₂ (**Figure S1**). TEM images show that the particle size of Pd and Ru NPs is ca. 4 nm and 2 nm, respectively. However, Pd/TiO₂ and Ru/TiO₂ were inactive in BA hydrogenation using hexane as the solvent (**Table S1**). This may be due to the stronger adsorption of BA on Ru and Pd than on Pt as discussed in the manuscript. The above discussions have been added in the revised manuscript.

Figure S1. TEM images of (a) Pd/TiO₂ and (b) Ru/TiO₂.

Table S1. The catalytic results of Ru/TiO₂ and Pd/TiO₂ in BA hydrogenation.

Cat.	Conv. (%)	Sel. (%)
Ru/TiO ₂	~0	--
Pd/TiO ₂	~0	--

Reaction conditions: 40 °C, 10 bar H₂, S/C = 700, 3 mL n-hexane, 2 h.

Question 3: “The dispersion results of Pt calculated by H₂ and CO chemisorption are quite different. Please check and specify.”

Response: Thanks for your nice reminding. The H₂ chemisorption was measured by H₂-O₂ titration method and this was clarified in the revised manuscript.

Question 4: “The drawing and layout of the diagrams are not standardized in this article, such as Figure 2.”

Response: Thanks for your good suggest. All the drawing and layout of the diagrams has been standardized in the revised manuscript.

Question 5: “The reaction temperature should be unified when calculating the reaction orders of the reactants.”

Response: Thanks for your good suggestion. The reaction orders of the reactants were re-performed at 60 °C in the revised manuscript.

Question 6: “Do the structure and composition of the catalysts change during the reaction? So it’s necessary to provide more characterization of the catalyst after the reaction.”

Response: Thanks for your good suggestion. The Pt/TiO₂ after 5 cycles was characterized by TEM and XPS techniques. The results show that used Pt/TiO₂ has similar particle size and electric structure as the fresh one (**Figure S9**), implying no obvious change in structure and composition during recycling process. The above discussion has been added in the revised manuscript.

Figure S9. (a) TEM image of Pt/TiO₂ after 5th recycle. (b) XPS spectra of Pt 4f of (i) Pt/TiO₂ and (ii) Pt/TiO₂ after 5th recycle.

Question 7: “There are many spelling and grammar mistakes in the article, which need to be thoroughly corrected. The references also have the problem of repeated citations.”

Response: The English has been polished in the revised manuscript.

Responses to Reviewer #2

Comments:

This ms reports on a study on benzoic acid hydrogenation on Pt/TiO₂ catalysts. The reaction is difficult and the catalytic performances are very good compared to benchmark catalysts.

I have read this contribution with great interest and have considered it for publication, but further evidences are still required to strengthen the conclusions.

Response: Thank you so much for your careful review and valuable suggestions. The questions are answered point-by-point as follows:

Question 1: “Hydrogenation reactions in multi-phase reactors can often be mass transfer limited and I find the argument presented lines 220-222 unsatisfactory. There are well-established protocols for discerning the role of intra-phase and interphase mass transfer limitations in catalytic systems (e.g., Madon-Boudart, Weisz Prater, Mears, among others). If the authors can clearly discern that the effects shown (lower E_a) represent kinetic events involving bond forming/bond cleavage and not physical rate processes involving mass transfer then the drawn conclusion can be made rigorously.”

Response: Thanks for your good suggest. A control experiment was carried out at different stirring rates and the results showed that the reaction rate remained constant in the range of 750–1200 rpm (**Figure 1**). The stirring rate for BA hydrogenation was set at 1000 rpm in order to exclude the influence of mass transfer.

Figure 1. Effect of the change in the stirrer speed (0 to 1200 rpm) on the reaction activity of the benzoic acid hydrogenation (40 °C, 10 bar H₂, S/C = 250).

Furthermore, the widely used criteria of mass transfer tests in three phases stirring reactors were used¹. The criteria, which define the ratio of observed rate to the maximum rate, are shown as followed:

$$\alpha_1 = \frac{r_A}{k_L a_B C_A^*} < 0.1 \quad \text{Eq. 1}$$

$$\alpha_2 = \frac{r_A}{k_s a_p C_A^*} < 0.1 \quad \text{Eq. 2}$$

Where α_1 describes the gas-liquid mass transfer ratio, and α_2 describes the liquid-solid mass transfer ratio, r_A is the initial reaction rate of BA hydrogenation (0.213 mol L⁻¹ h⁻¹), $k_L a_B$ is the gas-liquid mass transfer coefficient (Eq. 3), C_A^* is the saturation solubility in hexane that can be obtained from ref. 2 (0.075 kmol m⁻³, 298.15 K, 1.0 MPa H₂), k_s is the effective diffusivity (Eq. 4), and a_p is the catalyst external surface area (29 m² g⁻¹). If each requirement is met ($\alpha_1 < 0.1$, $\alpha_2 < 0.1$), it can be said that the system is free of gas-liquid and liquid-solid mass transfer.

The $k_L a_B$ in Eq. 1 can be calculated as follows³:

$$k_L a_B = 1.48 \times 10^{-3} (N)^{2.18} \left(\frac{V_g}{V_L} \right)^{1.88} \left(\frac{d_I}{d_T} \right)^{2.16} \left(\frac{h_1}{h_2} \right)^{1.16} \quad \text{Eq. 3}$$

The values used here are: N, the stirring speed, is 16.7 Hz; V_g , the volume of H₂, is 3 × 10⁻⁴ m³; V_L , the volume of liquid, is 3 × 10⁻⁴ m³; d_I , the diameter of the impeller, is 0.01 m; d_T , the diameter of the reactor tank, 0.015 m; h_1 , the height of the impeller from the bottom of the tank, is 0.003 m; h_2 , the height of the liquid in the tank, is 0.02 m. Thus, the $k_L a_B$ value is 0.03 s⁻¹. Applying r_A , $k_L a_B$ and C_A^* in Eq. 1 gives α_1 of 0.02, which is well below 0.1, indicating the system is not limited by the rate of gas to liquid mass transfer.

The liquid-solid mass transfer coefficient (k_s) in Eq. 2 can be calculated from the Sherwood number (Sh) in Eq. 4 (ref. 4,5):

$$Sh = \left[2 + 0.4 \left(\frac{\varepsilon d_p^4}{\nu^3} \right)^{\frac{1}{4}} Sc^{\frac{1}{3}} \right] \Phi_c = \frac{k_s d_p}{D_A} \quad \text{Eq. 4}$$

The right side of Eq.4: d_p , the specific surface diameter, can be obtained from the density (4.26 × 10⁶ g m⁻³) and the BET surface area of rutile (29 m² g⁻¹)^{4,5}. The calculated value of d_p is 4.86 × 10⁻⁸ m. D_A is the diffusivity of hydrogen in hexane. The experimental value of D_A is 62.38 × 10⁻⁹ m² s⁻¹ (298K)⁶.

The left side of Eq.4: ν is the kinematic viscosity of hexane. The experimental value of ν is 4.247 × 10⁻⁷ m² s⁻¹ (313K). The Schmidt number (Sc) is can be calculated from the ratio of $\nu/D_A = 6.8$ (ref. 7). Φ_c is the Carman's surface factor, which can be calculated from the d_p/d_p' . The approximate value of d_p' (the screen diameter) is 2 d_p (refs. 4,5). Thus, $\Phi_c \approx 0.5$. ε is the rate of flow energy supply per unit mass of liquid, which can be calculated from Eq. 5.

$$\varepsilon = \frac{N_p l^5 n^3}{V_L} \quad \text{Eq. 5}$$

where N_p is the impeller power number. $N_p = 3$ for the reaction system⁸. l is the diameter of impeller (0.03 m), n is the rps of impeller (105 rad s⁻¹), and V_L is the volume of liquid (3 × 10⁻⁴ m³). Eq. 5 gives $\varepsilon = 281$ m² s⁻³. Combining the above data

and Eq. 4, k_s with the value of 2.58 m s^{-1} can be calculated. From Eq. 2, a_p is the external surface area of the catalyst per unit volume of reactor and is calculated to be $1.03 \times 10^6 \text{ m}^2 \text{ m}^{-3}$. Using these values obtained and plugging into Eq. 2 yields an $\alpha_2 = 2.9 \times 10^{-4}$, which is orders of magnitude below 0.1 indicating the system is not mass transfer limited in the liquid-solid regime.

The above results show the kinetic data were in the absence of external mass transfer limitation. Finally, the low surface area and the extremely low pore volume of the TiO_2 diminish the presence of intra-particle mass transfer limitation.

We have added the above results in the section 1 of SI.

Question 2: “I also find the explanation given on the greater ease in activating hydrogen on an electron-deficient metallic species surprising. This goes against what is generally accepted: see for example the extreme case of single atom catalysts very deficient in electrons, which are only able to activate H_2 via heterolytic activation involving the support.”

Response: We agree with the Reviewer that H_2 activation on single atom catalysts is not easy. This cannot be assigned to the electron deficient metal sites. The Ru(II), Ir(III), Pd(II) metal complexes generally used in homogeneous catalysis could efficiently activate H_2 via heterolytic activation of rout. The low H_2 activation ability of single atom is mainly due to low efficiency of surrounding atoms in helping activation of H_2 . The electron-deficient Pt surface is more active in H_2 dissociation is possibly related with enhanced H_2 adsorption on electron deficient Pt surface, which is consistent with the previous results (Panagiotopoulou, P., et al J. Catal. 2008, 260, 141–149.).

Question: “If the activation of the carboxylic function can contribute to increasing the rate of hydrogenation, I do not think that it is sufficient to explain the observed differences. Indeed, esters are also hydrogenated very quickly (Table 3). It would certainly be relevant to study the possibility of H-spillover (for example by H_2 -TPD/MS) on the two catalysts Pt/ TiO_2 and Pt/ TiO_2 -450. In fact, the reduction of rutile TiO_2 at $450 \text{ }^\circ\text{C}$ must deeply modify the surface of the rutile support (oxygen vacancy formation etc...). H_2 spillover enhanced hydrogenation capability of Pt/ TiO_2 catalyst could be an explanation of the observed higher activity of the Pt/ TiO_2 catalyst.”

Response: We agree with the Reviewer that H_2 spillover may enhance the hydrogenation activity of Pt/ TiO_2 . H_2 -TPD measurement was performed to reveal the hydrogen species formed on Pt/ TiO_2 and Pt/ TiO_2 -450 (**Figure S8**). For Pt/ TiO_2 , the low-temperature peak can be assigned to desorption of hydrogen on metallic Pt and the peaks above $200 \text{ }^\circ\text{C}$ are related to hydrogen species on TiO_2 derived from hydrogen spillover^{30, 51}. However, only two broad and weak peaks were observed for Pt/ TiO_2 -450, showing the difficulty in H_2 spillover possibly due to the

dehydroxylation of TiO₂ during heat treatment³⁰. The easier H₂ spillover on Pt/TiO₂ may also contribute to the high activity of Pt/TiO₂. The above discussions were added in the revised manuscript.

Figure S8. H₂-TPD profiles of (i) Pt/TiO₂ and (ii) Pt/TiO₂-450

Responses to Reviewer #3

The authors have nicely presented the work entitled “Pt/TiO₂ Catalyzed Hydrogenation of Benzoic Acid with Unprecedented High Activity” where they have claimed very high activity of Pt/TiO₂ for the conversion of Benzoic acid (BA), terephthalic acid, iso-phthalic acid, trimesic acid and other BA derivatives to corresponding aromatic saturated products. The whole work is very well designed and performed.

The present manuscript may be considered for publication in the Communications Chemistry journal after consideration of suggested revisions.

Response: Thank you so much for your careful review and valuable suggestions. The questions are answered point-by-point as follows:

Question 1: “The author should relook the title, needs modification. They should avoid the terminology such as “Unprecedented High Activity” in the title. The author also mentioned “benzoic acid” in title but in substrates they have used some other aryl substituted alkyl acids, may consider in title and make it general.”

Response: Thanks for your good suggest. A more general title “Hydrogenation of Benzoic Acid Derivates over Pt/TiO₂ under Mild Conditions” was used in the revised manuscript.

Question 1: “The scale bar size (i.e. xyz nm) should also have to be written on HRSEM and HAADF-STEM images in Figure 2 of the manuscript and Figure S1-

Figure S3 and Figure S6 of supporting information for better data communication to readers.”

Response: Thank you for your good suggest. The scale bar has already been added in the HRSEM, HRTEM and TEM images (Figure 2, Figure S1-Figure S4 and Figure S9).

Question 2: “The Selective Area Electron Diffraction (SAED) and X-Ray Diffraction (XRD) characterization of Pt/TiO₂ NPs should also be included in the manuscript for the confirmation of active planes responsible for desired transformation.”

Response: It is hard to obtain a good diffraction spots in the SAED pattern due to the ultrasmall size of Pt NPs. However, the crystal plane of Pt (111) with plane spacing of 0.23 nm could be seen clearly in the HRTEM image. We have added the XRD and HRTEM image in the revised manuscript.

XRD patterns of (a) Pt/TiO₂ and (b) Pt/TiO₂-450.

HRTEM image of Pt/TiO₂ (The inset is the plane spacing of Pt NPs)

Question 3: “The authors should also provide the hot filtration test and Inductively coupled plasma atomic emission spectroscopy (ICP-AES) analysis details of Pt/TiO₂ catalyst for clarification regarding true heterogeneous nature of the catalyst and Pt metal leaching during the course of reaction respectively.”

Response: Thanks for your nice suggestion. The hot filtration reaction was performed in the revised manuscript. The Pt/TiO₂ was removed from the reaction system after BA conversion reaching to 50%. Then the filtrate was recharged with 10 bar H₂. After 48 min, no further increase in conversion was observed (**Figure S6**). Meanwhile, the concentrations of Pt in the reaction solution were below the detection limit of ICP-AES, confirming the heterogeneous nature of Pt/TiO₂. These discussions have been added in the revised manuscript.

Question 4: “In Page No. 18, Line No. 352, ‘NaBH₄ as the reactant’ should be replaced with ‘NaBH₄ as the reductant’.”

Response: Thanks for the nice reminder. The mistake has been corrected.

Question 5: “Authors have cited two research papers (Ref. 34, 58) published in 2018-2019 approaching Pt metal deposition precipitation method using NaBH₄ as the reductant (Page No. 18, Line No. 352) whereas, a work published by Das et al. using NaBH₄ as the reductant for solid supported Pt NPs preparation through deposition precipitation method in 2013 (Green Chem., 2013,15, 3421-3428) should be considered in the text and reference.”

Response: Thank you for the good suggestion. The reference (Green Chem., 2013,15, 3421-3428.) have been added as ref 59 in the revised manuscript.

Question 6: “Although authors have very well explained the role of carboxylic group on phenyl ring for hydrogenation reaction. However, have the authors tried other functional groups such as aldehydes, ketones and alcohols for this protocol. If yes, then they should mention the results in one line.”

Response: As the Reviewer suggested, the catalytic performance of Pt/TiO₂ was also tested in hydrogenation of benzaldehyde, benzylalcohol, and acetophenone (Tables 1, 2). The reaction pathway of aromatic ketones/aldehydes hydrogenation is very complicated and many products co-exist (Scheme 1). Generally, they involve a series of hydrogenolysis and hydrogenation process (Jayakumar, S. Chem. Eur. J., 23, 7791-7797 (2017). Chen, M. ACS Catal., 2, 2007–2013 (2012).). Though Pt/TiO₂ shows high activity but the selectivity is poor. Furthermore, aromatic aldehydes/ketones do not belong the BA derivatives, so we did not mention this in the revised manuscript.

Scheme 1. Possible reaction pathways of benzaldehyde hydrogenation.

Table 1. The product distribution for benzaldehyde/benzylethanol hydrogenation over Pt/TiO₂.

Substrate					Benzaldehyde	~0	17%	~0	56%	27%
Benzylethanol	--	16%	~0	57%	27%

Reaction conditions: 40 °C, 10 bar H₂, 0.15 mmol of substrate, S/C = 250, 3 mL n-hexane, 1 h for benzaldehyde, 30 min for benzylalcohol. Conversion > 99%.

Table 2. The products distribution of acetophenone hydrogenation over Pt/TiO₂.

Substrate					Acetophenone	38%	5%	39%	9%	9%

Reaction conditions: 40 °C, 10 bar H₂, 0.15 mmol of substrate, S/C = 250, 3 mL n-hexane, 1 h. Conversion > 99%.

REVIEWERS' COMMENTS:

Reviewer #1 (Remarks to the Author):

This paper has been well revised and I recommend the publication.

Reviewer #2 (Remarks to the Author):

The authors have addressed all of the Reviewer's concerns, including additional data, broadening the discussion, or provided satisfactory responses for not taking action. I think this article will be of interest to the catalysis community and therefore recommend its acceptance.

Reviewer #3 (Remarks to the Author):

The manuscript has been thoroughly revised by the authors in accordance to the suggestion and now it can be accepted for the publication in Communications Chemistry journal.